# Dynamic Structural Changes in a Population of *Rhododendron huadingense*, a Rare and Endemic Species in Zhejiang, East China

**DOI:** 10.3390/plants14223406

**Published:** 2025-11-07

**Authors:** Ke Hao, Anguo He, Shuran He, Weijie Chen, Zilin Chen, Xin Cai, Pan Wang, Yu Chen, Yifei Lu, Xiaofeng Jin

**Affiliations:** 1College of Forestry and Biotechnology, Zhejiang A&F University, Hangzhou 311300, China; 2Administration of Zhejiang Dapanshan National Nature Reserve, Doctoral Innovation Workstation of Dapanshan Museum, Jinhua 322300, China; 3Traditional Chinese Medicine Industry Development and Promotion Center of Pan’an County, Jinhua 322300, China; 4College of Life and Environment Sciences, Hangzhou Normal University, Hangzhou 311121, China

**Keywords:** *Rhododendron huadingense*, age structure, static life table, survivorship curves, population structure, population dynamics

## Abstract

*Rhododendron huadingense* is a nationally protected wild plant species in China with a small population and narrow natural distribution, which is experiencing severe anthropogenic threats. The demographic structure and trends of *R. huadingense* on Mount Dapan in Zhejiang Province were analyzed to help researchers understand its population dynamics by using static life tables, quantitative dynamic indices, survivorship curves, and survival analysis based on three censuses of a 1 ha plot in 2012, 2017, and 2022. Over this decade, the population of *R. huadingense* declined by 9.58% from 668 to 604 individuals. From 2012 to 2022, the diameter class structure of the population consistently exhibited a pyramidal pattern, with the number of individuals initially increasing and then decreasing as diameter classes advanced. Over this decade, the diameter class structure of the population consistently showed a declining trend. Dynamic indices showed that the quantitative dynamic index of the population without external interferences was greater than with external interferences, and that both were greater than zero, suggesting growth potential. The maximum risk probability in response to random disturbance from 2012 to 2022 (2017 > 2022 > 2012) was greater than zero but relatively small, indicating underlying demographic instability. Life tables showed Deevey Type II survivorship with mortality rates decreasing in larger diameter classes. These demographic patterns indicate that *R. huadingense* is transitioning from recruitment-driven growth to senescence-dominated population dynamics. Urgent conservation interventions are needed, including (1) enhancing seedling establishment, (2) improving habitat quality, (3) promoting natural regeneration processes.

## 1. Introduction

Population structure and dynamics are fundamental components of population ecology, determining species persistence across spatiotemporal scales [1,2]. The diameter structure of a population, as the most basic characteristic of population demography, reflects historical population processes and provides critical insights for predicting future demographic trajectories under changing environmental conditions [3,4]. For rare and endemic species, understanding population dynamics—spatiotemporal fluctuations in abundance and rates of demographic change [5,6]—becomes particularly important, because these taxa exhibit high extinction vulnerability to demographic stochasticity and environmental disturbances due to their small population sizes, restricted distributions, and limited dispersal capabilities [7,8,9]. A comprehensive analysis of population structure and dynamics can reveal demographic bottlenecks, recruitment limitations, and life-stage-specific vulnerabilities that directly influence extinction risk in endemic populations. These demographic insights provide important information for assessing population viability and developing targeted conservation strategies, particularly for geographically restricted species confronting accelerating habitat degradation and climate change [10,11].

The mountainous regions of East China harbor numerous endemic species; preserving these species will require this type of demographic assessment and targeted conservation strategies, which are urgently needed [12,13,14]. *Rhododendron huadingense* is a deciduous shrub 1–4 m tall, bark gray-brown with prominent longitudinal fissures, leaves membranous to thinly chartaceous, 4–5-verticillate at branch apex, blade elliptic, margin finely serrulate with teeth gland-tipped or shortly ciliate, inflorescence a terminal umbel-like raceme, 2–4-flowered, corolla funnel-campanulate, pale purplish-red to purplish-red. It exhibits hysteranthous flowering with leaf buds undeveloped at anthesis, presenting an elegant and archaic appearance with high ornamental value [15,16]. Since its initial discovery by Professors Bingyang Ding and Yunyi Fang in 1990 on Mount Huading in Tiantai County, *R. huadingense* has been recorded at only a few scattered locations in the following locations within county-level cities, including Mount Siming in Yuyao, Mount Beishan in Wucheng, Mount Dapan in Pan’an, Mount Kuocang in Linhai, and in adjacent regions within Zhejiang. *R. huadingense* inhabits *Pinus taiwanensis* forests as well as coniferous and broad-leaved mixed forests above 700 m elevation [13,17]. This species represents a significant aspect of community ecology and plays an essential role in mountainous forest ecosystems. The structural and functional stability of forest ecosystems is significantly impacted by this species [18,19]. This species is a member of a phylogenetically distinct lineage within genus *Rhododendron*, which suggests that it should be classified into a new subgenus harboring unique genetic resources crucial for understanding the biogeographic evolution of the East Asian *Rhododendron* flora [15,20].

Given these multiple threats, along with its phylogenetic uniqueness and ecological value, *R. huadingense* has been designated as a National Category II Protected Wild Plant since 2021, with additional conservation status as a Zhejiang Provincial Key Protected Species and an Extremely Small Wild Population since 2012 [21,22,23]. *R. huadingense* possesses exceptionally high conservation value. Its habitat represents the sole distribution area for this endemic species and also supports a sympatric population of multiple threatened plant species. Collectively, they form a unique plant community in the region and play a critical role in maintaining regional plant diversity [13,16]. Moreover, its endangered status designates it as a keystone species for conserving plant endemism in Zhejiang, this species is crucial for maintaining regional phytogeographic distinctiveness, counteracting floristic homogenization, and mitigating extinction risks [15,18]. Many scholars have conducted preliminary studies on *R. huadingense*, with research efforts primarily focused on areas such as resource surveys, community characteristics, genetic diversity, and artificial propagation techniques [15,18,24,25,26]. However, existing studies have largely emphasized static descriptions of population characteristics and explorations of conservation techniques, while systematic monitoring and investigation of the dynamic patterns and driving mechanisms of its wild populations across different successional stages or over long-term temporal scales remain notably lacking.

According to MaxEnt model predictions, the contemporary potentially suitable habitat of *R. huadingense* is primarily distributed in specific high-elevation areas of Zhejiang and Anhui Provinces. However, its actual current distribution is confined to mid-to-high-elevation mountainous regions in east-central Zhejiang [27]. Based on field investigations of wild populations in 2012, the Mount Dapan National Nature Reserve in central Zhejiang harbors the most significant population of *R. huadingense*; this population has experienced minimal anthropogenic disturbance and benefits from having a very suitable habitat. This population provides a critical scientific basis for reintroduction and ex situ conservation efforts for *R. huadingense* [18]. Therefore, this study focuses on the rare and endangered species *R. huadingense*. Field investigations were conducted to examine the population regeneration characteristics and distribution patterns within its natural range. By analyzing the age structure, static life table, and survival curve of wild populations, the study aims to address the following scientific research aspects: (1) the survival status of the population over the past decade; (2) the dynamic trends of the population; (3) the mechanisms underlying its endangerment and conservation. The findings will elucidate the population maintenance mechanisms and endangerment causes from an ecological perspective, while providing a scientific basis for developing effective conservation strategies and enabling targeted protection of this species.

## 2. Results

### 2.1. Stand-Scale Distribution Patterns and Diameter Class Structure of Rhododendron huadingense

Field surveys have documented persistent declines in *R. huadingense* abundance across all census intervals, with individual counts falling from 668 in 2012 to 678 in 2017 and 604 in 2022. From 2012 to 2017, the population increased by 10 individuals, while from 2017 to 2022, it decreased by 74 individuals. The diameter class structure of the *R. huadingense* population surveyed in 2012, 2017, and 2022 all exhibited a pyramidal pattern (Figure 1). The number of individuals in the population initially increased and then decreased with advancing diameter classes. The peak number of individuals consistently occurred in Class IV, with Classes IV and V having the highest abundance. The number of individuals in Classes I and II was low. From 2012 to 2022, the number of individuals in Classes III to VI showed a declining trend, while the number of individuals in Classes VII to IX demonstrated an increasing trend over the decade. In the surveys conducted in 2012, 2017, and 2022, the *R. huadingense* individuals in Classes I and II were juveniles, accounting for a relatively low proportion of the total population: 7.93%, 11.21%, and 14.4%, respectively. Classes III to V represented saplings, with their proportion decreasing year by year, accounting for 70.66%, 62.24%, and 48.18% of the total population, respectively. Classes VI to VII were medium-sized trees, and their proportion increased over the years, representing 20.36%, 24.93%, and 33.44% of the total population, respectively. The number of old-aged trees was small, with zero individuals recorded in 2012, and only three individuals each in 2017 and 2022. The *R. huadingense* population was characterized by the highest abundance of saplings, a relatively sufficient number of middle-aged individuals, and a limited proportion of juveniles. The number of old-aged individuals remained very low. Therefore, the diameter class structure of the *R. huadingense* population indicates a declining overall population trend throughout the decade from 2012 to 2022.

### 2.2. Population Dynamics of Rhododendron huadingens

The inter-class dynamic indices (*V_n_*) between successive diameter classes showed variable patterns across the survey periods (Table 1). In both 2012 and 2017, the values of *V*_1_–*V*_3_ for the *Rhododendron huadingense* population were all less than 0, while *V*_4_–*V*_8_ were all greater than 0, indicating that the population exhibited a “decline–growth” structural dynamic relationship during these years. In 2022, *V*_1_–*V*_2_, *V*_4_–*V*_5_, and *V*_7_–*V*_8_ were greater than 0, while *V*_3_ and *V*_6_ were less than 0, suggesting that the population displayed a “growth–decline–growth–decline–growth” structural dynamic relationship in that year. Across the three surveys, the young individuals of the *R. huadingense* population in diameter classes I, II, and III showed a declining trend in both 2012 and 2017, while the middle and old individuals generally exhibited growth trends. In 2022, diameter classes III and VI showed a declining trend, while the other diameter classes displayed growth trends, which is consistent with the population’s age structure (Figure 1). The values of *V_pi_* and *V′_pi_* increased year by year from 2012 to 2022, with the overall quantitative dynamic index *V_pi_* (2022 > 2017 > 2012) > *V_pi_* (2017 > 2022 > 2012) > 0, indicating that the *R. huadingense* population exhibited a gradually increasing trend over the 10 year period from 2012 to 2022, regardless of disturbance. The presence of external disturbances slowed the population’s growth rate; notably, this slowing effect was particularly pronounced in 2022. The value of *V′_pi_* approached 0, suggesting that the population grew slowly under the influence of disturbances during the three surveys and gradually stabilized over time. The maximum risk probability in response to random disturbance (*P_max_*) from 2012 to 2022 (2017 > 2022 > 2012) was greater than 0 but relatively small. The population exhibits a certain sensitivity to stochastic disturbances. However, in 2012, the probability of the population being affected by disturbances was the lowest, indicating the highest stability during that year.

### 2.3. Static Life Table and Survival Rate Curve of Rhododendron huadingense

#### 2.3.1. Static Life Table Analysis

The static life table reflects the fundamental survival status of the *Rhododendron huadingense* population. As shown in Table 2, the *R. huadingense* population exhibited substantial variation in individual counts across diameter classes in all census years, generally decreasing with advancing diameter classes and exhibiting a year-by-year declining trend. Across all three censuses, the *R. huadingense* population exhibited a monotonic decline in standardized survivorship (*l_x_*) with increasing diameter class. Life expectancy (*E_x_*), quantifying the mean survival capacity within each cohort, overall showed a declining trend across the three surveys in 2012, 2017, and 2022. However, the life expectancy for each diameter class in 2022 was higher than that in 2012 and 2017, indicating an improvement in the average survival capacity of the population in 2022 compared to the earlier years. Across all three years, diameter class I exhibited the highest life expectancy, suggesting vigorous growth at this stage. As community increased, survival capacity and adaptability to the environment gradually declined, particularly after diameter class VI in 2012, 2017, and 2022. With individuals becoming overmature, their numbers decreased significantly, and the life expectancy of the population declined markedly, indicating that the population was gradually approaching its physiological lifespan.

#### 2.3.2. Mortality and Disappearance Rate Curve Analysis

The mortality rate (*Q_x_*) and disappearance rate (*K_x_*) curves of the *Rhododendron huadingense* population in 2012, 2017, and 2022 exhibited largely consistent trends (Figure 2). Both rates remained relatively stable with minor increases from diameter classes I to V, indicating that after environmental filtering, a small proportion of individuals failed to transition successfully to the next stage. Beyond diameter class VI, both mortality and disappearance rates increased rapidly with significant fluctuations, particularly in 2022 where variations were more pronounced. As environmental requirements intensified and interspecific competition became more severe, the number of surviving individuals declined markedly. Combined with the static life table and quantitative dynamic analysis, these results demonstrate that the number of individuals in diameter classes VII to IX sharply declined over the decade, indicating an unstable state in these stages of the population.

#### 2.3.3. Survivorship Curve Analysis

The survivorship curves for *Rhododendron huadingense* across the three census years exhibited consistent Deevey-II type patterns. All three curves showed gradual declines in standardized survivorship from diameter classes I to VI, followed by a sharp decrease in classes VI to IX (Figure 3). To quantitatively evaluate the curve shapes, we fitted both power and exponential functions to the data (Table 3). The fitting results demonstrated that for the period 2012–2022, the exponential function model exhibited higher *R*^2^ and F values, and lower *p* values, compared to the power function model. This indicates that the exponential function model provided a better fit, suggesting that the survival curves of the *R. huadingense* population in all three surveys align more closely with the Deevey-II type. The highest number of surviving individuals was observed in diameter class I, with a gradual decline as diameter classes advanced. The population remained relatively stable from diameter classes I to VI, exhibiting high survival rates. Beyond diameter class VII, the number of surviving individuals decreased sharply and remained at low levels, accompanied by increased mortality. These results indicate that the *R. huadingense* population has a limited number of young individuals and is in a state of decline. In 2022, the number of surviving individuals from diameter classes I to VI was lower than in 2012 and 2017, while beyond diameter class VI, the number of surviving individuals in 2022 was higher than in the previous surveys. This pattern is consistent with the observed diameter structure.

### 2.4. Survival Analysis of Rhododendron huadingense

The survival rate of the *Rhododendron huadingense* population over a 10-year period showed a monotonic decreasing trend with advancing diameter classes, while the cumulative mortality exhibited a monotonically increasing trend. The two are complementary, consistent with the biological characteristics of the species (Figure 4). The magnitude of change in survival and cumulative mortality is greater in the early stages than in the later stages. The changes are more pronounced in diameter classes I to VII, with the most significant changes occurring in classes V to VII. After diameter class VII, the magnitude of change tends to level off. Between 2012 and 2017, the survival and cumulative mortality rates were equal between diameter classes V and VI, indicating a balance in survival and mortality during this period. In 2022, this balance was reached at diameter class VI, after which the cumulative mortality (*F_i_*) exceeded the survival rate (*S_i_*). The population entered a decline later in 2022 compared to 2012 and 2017. Before diameter class VII, the rates of increase or decrease were more substantial. After diameter class VII, the changes in survival and cumulative mortality slowed, suggesting that the physiological functions of the *R. huadingense* population gradually enter a senescent phase beyond this diameter class, indicating a trend toward population decline.

From Figure 4b, it can be observed that the trends of the mortality density (*f_i_*) and the hazard rate (*λ_i_*) for the *R. huadingense* population in 2012, 2017, and 2022 are generally consistent. The mortality density (*f_i_*) shows relatively smooth changes in the early stages but significant fluctuations in the later stages. From 2012 to 2017, the changes are relatively smooth in diameter classes I to V, while considerable fluctuations occur in diameter classes V to IX, with a sharp increase in classes V to VII and a noticeable decline in classes VII to IX. In 2022, the fluctuations are less pronounced, indicating that the mortality rate of *R. huadingense* in diameter classes V to IX is unstable compared to 2012 and 2017 but relatively more stable in 2022.

Compared to the mortality density (*f_i_*) curve, the hazard rate (*λ_i_*) curve for the *R. huadingense* population in 2012, 2017, and 2022 (Figure 4b) exhibits a dynamic trend of “increase–decline.” The hazard rate steadily rises from diameter classes I to VI, peaks in class VI, and gradually declines from classes VI to IX. After class VI, as individuals gradually enter physiological senescence, their numbers progressively decrease. In 2012 and 2017, the hazard rate showed an upward trend in classes V to VII, while in 2022, the increase occurred in classes VI to VII. This indicates that as the *R. huadingense* population ages, limited resources lead to intense intraspecific competition, resulting in a sharper rise in the hazard rate function.

The survival rate of the *R. huadingense* population in diameter classes VI to IX is low, while the cumulative mortality, mortality density, and hazard rate are high, reflecting an unstable vital state. Overall, the population in 2022 was more stable compared to 2012 and 2017. Analysis using the four functions reveals that the *R. huadingense* population experiences smooth changes in the early and middle stages but significant fluctuations in the later stages, gradually declining over time. These characteristics align with the findings from the survival curve, population mortality rate, and mortality rate curve analyses.

## 3. Discussion

### 3.1. Diameter-Class Structure of a Rhododendron huadingense Population

The diameter-class structure of a population can serve as a proxy for population age structure to assess both habitat adaptation and population viability [27,28,29]. The diameter-class structure of the *Rhododendorn huadingense* population in Mount Dapan displayed a pyramidal distribution both in 2012, 2017, and 2022. The population exhibited a marked decline in sapling abundance accompanied by proportional increases across older diameter classes, resulting in critically insufficient recruitment at early ontogenetic stages, showing a declining trend in the total population. These findings demonstrate significant congruence with the population structure and dynamics of *Abies fargesii* Rehd—*Rhododendron simsii* Planch at the alpine treeline in the high-elevation Gannan mountains [30].

Analysis of the dynamic indices (*V_pi_* > *V′_pi_* > 0) for the *R. huadingense* population from 2012 to 2022 reveals that while the population exhibited an expanding structure over the decade, the perturbation-responsive dynamic index (*V′_pi_*) asymptotically approached zero. The annual increase in the extreme value of the maximum risk probability in response to random disturbance (*P_max_*) from 2012 to 2022—though remaining marginally above zero—suggests population instability and a weak growth trend under external stochastic disturbances. These patterns show remarkable consistency with findings by Jiang et al. [31] in their structural and dynamic analysis of a *Rhododendron feddei* population. The population structure of *R. huadingense* remained relatively stable overall but exhibited signs of transitioning toward a declining trend. While the current population is adapted to its existing habitat, the scarcity of saplings raises concerns about the long-term viability of this population. Without sufficient juvenile recruitment, the long-term viability of this population will, over time, become increasingly uncertain.

The population decline and aging structure observed in this population of *R. huadingense* can be attributed to a combination of intrinsic biological constraints and external pressures. Its strict niche specialization—dependence on high-elevation sunny slopes with poor soils—limits its adaptability to climate change and habitat degradation. Reproductive challenges include inefficient seed dispersal and difficult seedling establishment resulting from competition for light, the presence of thick litter layers, and intense resource competition with understory plants [25,32]. Anthropogenic disturbances such as habitat fragmentation, and illegal plant collection further reduce the availability of suitable habitats and population connectivity, respectively [32]. Interspecific competition with other woody plants and dense bamboo understories also restricts regeneration [26]. Demographically, the population exhibits a declining age structure with an insufficient number of juvenile plants, which is likely to lead to reduced genetic diversity, inbreeding depression, and diminished adaptive potential [33].

### 3.2. The Dynamic Trends of the Rhododendron huadingense Population

Static life tables, survival curves, and survival function analysis can reflect a population’s survival status and environmental impact degree and predict its future trends [34,35,36]. In this study, static life tables from three surveys show that the standardized survival number (*l_x_*) and life expectancy (*E_x_*) of the studied *Rhododendron huadingense* population decline with increasing diameter classes and indicate that the population is approaching its physiological lifespan. This pattern aligns consistently with the species’ biological characteristics. The mortality rate (*Q_x_*) and disappearance rate (*K_x_*) curves of the *R. huadingense* population showed generally consistent trends, both exhibiting a continuous increase with diameter class and reaching their maximum values in diameter classes VI–IX. Both life expectancy and survival rate showed significant declines after diameter class VI, VII, suggesting that increasing diameter class leads to growth resources demands and intensified competition among individuals. These findings are consistent with Wang’s analysis of community structure and population dynamics in a *Rhododendron xishuiense* population from Guizhou [37].

The survival curves of the *R. huadingense* population in 2012, 2017, and 2022 tended to follow the Deevey-II pattern. However, due to the continuous decline in the number of juvenile individuals and scarcity of mature trees in the study area, if the survival rate of juvenile and sapling individuals cannot be improved, the middle-age-class individuals will not be adequately replenished over time, potentially leading the population toward extinction. The survival rate and mortality rate of the population were inversely related. As the diameter class increased, the survival rate declined while the cumulative mortality rate gradually rose. The mortality density functions for 2012, 2017, and 2022 fluctuated, while hazard rate curves rose initially then declined, indicating substantial environmental influence on the *R. huadingense* population. The *R. huadingense* population studied here exhibits slow growth. Seedlings and saplings (low-diameter classes) show vigorous vitality and require relatively few resources, resulting in low mortality rates. As the plants mature, their demand for space and resources increases. The authors believe that intense competitive pressure and environmental filtering lead to a sharp rise in mortality. Upon entering the mid-to-late adult stage, intraspecific competition weakens somewhat as a result of prior self-thinning, reducing the mortality risk. After diameter class VI, both mortality and disappearance rates increase significantly, likely a result of physiological senescence, declining ability to compete, and reduced environmental adaptability [8]. These findings align with similar conclusions reported by Jiang, Jin, and Jia [38,39,40] in studies of a population of the endangered species *Syringa pinnatifolia*, *Rhododendron chrysanthum*, and *Fraxinus sogdiana*.

The decline of the studied *R. huadingense* population poses the potential existence of serious ecological and species-specific consequences resulting from the population decline. Ecologically, the decline reduces biodiversity by disrupting food web relationships—such as diminishing insect populations that rely on the plant—and impairs ecosystem functions including soil stabilization and nutrient cycling, accelerating erosion and altering material flow. For the species itself, population decline leads to reduced genetic diversity, increased inbreeding, and accumulation of deleterious alleles, thereby weakening adaptive potential [41,42]. Additionally, diminished reproductive capacity—fewer mature individuals result in lower pollination success and recruitment—further exacerbates the decline. The lack of juvenile individuals also reduces resilience to environmental changes such as shifts in climate and more significant impacts resulting from natural disasters or novel diseases, increasing extinction risk [43].

### 3.3. Conservation Implication for the Rhododendron huadingense

The proposed conservation strategies for the *Rhododendron huadingense* population on Mount Dapan include enhancing protection of this species, preventing anthropogenic disturbance, and simultaneous emphasizing both individual plant protection and protection of both their communities and habitats. Appropriate human intervention should be implemented, such as managing the forest understory to promote seed germination and the growth of seedlings into saplings [32,35,44,45]. These integrated approaches aim to create favorable conditions for seedling regeneration and maintaining the stability of the host plant community. Based on the biological characteristics of *R. huadingense*, population restoration should be conducted through artificial breeding methods in native habitats or ecologically similar environments to increase population abundance. Seedling propagation and reintroduction programs are being carried out to enhance seedling survival rate and quality, optimize population structure allocation, and achieve the goal of healthy population development.

## 4. Materials and Methods

### 4.1. Study Area

Zhejiang Mount Dapan National Nature Reserve is located in Pan’an County, Zhejiang Province (28°57′05″–29°01′58″ N, 120°28′05″–120°33′40″ E), situated within the central branch of the Zhejiang Mountainous Range. The reserve covers an area of 4558 ha, with its highest peak, the summit of Mount Dapan, reaching an elevation of 1245 m. The region experiences a subtropical monsoon climate characterized by distinct seasons, moderate annual temperatures, abundant sunshine, ample rainfall, and high humidity. Precipitation and temperature patterns show strong seasonal synchronization, with warm, wet summers and cool, dry winters. The mean annual average temperature is 15.0 °C, with recorded temperature extremes ranging from 36.9 °C to −9.5 °C. Mean average annual precipitation is 1427.8 mm, and the average annual sunshine duration reaches 1827.6 h [15].

### 4.2. Plot Establishment and Population Survey

Referring to the technical specifications for the establishment and monitoring of large plots produced by the Center for Tropical Forest Science and the Chinese Forest Biodiversity Monitoring Network [46], a 1 ha permanent dynamic monitoring plot for *Rhododendron huadingense* was established in July 2012 within the *R. huadingense* forest of the Mount Dapan National Nature Reserve. The plot was subdivided using the contiguous grid method. During the surveying process, marker points were set every 20 m using polyvinyl chloride pipes as markers, dividing the plot into 25 quadrats of 20 m × 20 m each. Each basepoint polyvinyl chloride pipe was numbered, with row and column codes combined to form identify each quadrat. To facilitate field investigations, each 20 m × 20 m quadrat was further divided into four 10 m × 10 m sub-quadrats and sixteen 5 m × 5 m sub-sub-quadrats. Re-surveys were conducted every five years. The plot had a minimum and maximum elevations of 1115 and 1145 m, respectively. Each entire plot was square in shape, with the southwestern corner designated as the origin. The east–west and north–south directions correspond to the x- and y-axes (each 100 m in length). Using a total station, the entire plot was divided into 100 quadrats of 10 m × 10 m, and the elevation of each quadrat was measured to create a topographic map (Figure 5).

A population survey was conducted in the *R. huadingense* sampled plot of Mount Dapan. For woody plants with a Diameter at Breast Height (DBH) ≥ 1 cm, the DBH was measured at 1.3 m above ground and marked with paint. Newly encountered individuals were measured at the same height, then tagged and painted. Additional data recorded for each individual included species identification, tree height, crown width, and global positioning system coordinates. Supplementary records were made for unlabeled individuals of *R. huadingense*.

### 4.3. Data Analysis

#### 4.3.1. Diameter-Class Structure of *Rhododendron huadingense* Population

Because the distributions of the age and diameter class structure of various tree species exhibit consistent patterns under similar environmental conditions [47], we employed the diameter-based age class substitution method to analyze population dynamics by classifying individuals into discrete diameter classes. Following the diameter class classification system established for woody plants on Mount Baishanzu, Zhejiang Province [48], each individual of *Rhododendron huadingense* was categorized into one of seven diameter classes as follows: Class I (1 cm ≤ DBH < 2 cm), Class II (2 cm ≤ DBH < 3 cm), Class III (3 cm ≤ DBH < 4 cm), Class IV (4 cm ≤ DBH < 5 cm), Class V (5 cm ≤ DBH < 7 cm), Class VI (7 cm ≤ DBH < 10 cm), and Class VII (DBH ≥ 10 cm). The number of individuals in each class was quantified to characterize the population diameter structure.

#### 4.3.2. Quantification of Population Dynamics

To address the limitations of traditional diameter class classification methods and enable more accurate analysis, this study adopted methodology developed by Chen in 1998, which integrates curve-fitting functions and quantitative indices for analyzing plant population structure [34,49]. The formula is as follows:(1)$$V_{n}=\frac{s_{n}-s_{n+1}}{max(s_{n},\;s_{n+1})}\times 100\%$$(2)$$V_{pi}=\frac{1}{\sum_{n=1}^{K-1}S_{n}}\cdot \sum_{n=1}^{K-1}\left(S_{n}\cdot V_{n}\right)$$
where *V_n_* and *V_pi_* represent the inter-class and intrinsic dynamic indices, which quantify the variation in population size between two consecutive diameter classes and across the complete size structure under disturbance-free conditions, respectively; *S_n_* and *S_n_*
_+ 1_ represent the number of individuals in diameter classes *n* and *n* + 1, respectively.

When accounting for potential future disturbances, the composite dynamic index of population size variation becomes dependent upon both the cohort-specific population abundances (*S_n_*) and the total number of age structures (*K*):(3)$${V^{\prime }}_{pi}=\frac{\sum_{n=1}^{K-1}\left(S_{n}\cdot V_{n}\right)}{K\cdot \min(S_{1},S_{2},S_{3}\cdots S_{k})\cdot \sum_{n=1}^{K-1}S_{n}}$$(4)$$P_{max}=\frac{1}{K\cdot min\left(S_{1},S_{2},S_{3}\cdots S_{k}\right)}$$
where *K* denotes the total number of age classes in the population, and *S_n_* represents the number of individuals in the *n*-th age class. *V′_pi_* represents the composite dynamic index of the variation in population size variation that incorporates potential future disturbances; *P_max_* denotes the maximum risk probability of population decline under external environmental disturbances; the values of *V_n_*, *V_pi_*, and *V′_pi_* (positive, zero, or negative) indicate the growth, stability, and decline between adjacent diameter classes, respectively.

#### 4.3.3. Static Life Table and Survival Curve of *Rhododendron huadingense*

Static life tables can effectively characterize the mortality and survival processes of perennial plant populations, making them a valuable tool for analyzing plant population dynamics [50]. During raw data processing, smoothing techniques can be applied to prevent the occurrence of negative mortality rates [35,50]. We thus constructed static life tables for *Rhododendron huadingense* based on fundamental data including the distribution of the diameter class structure and individual counts per diameter class [51]. The parameters and their corresponding computational formulas are specified as follows: *X*, diameter class (used as a proxy for diameter class); *N_x_*: number of surviving individuals per size class;(5)$$l_{x}=a_{x}/a_{0}\;\times \;1000$$*L_x_* = (*N_x_* + *N_x_* + 1)/2(6)*D_x_* = *N_x_* − *N_x_* + 1;(7)*Q_x_* = *D_x_*/*N_x_*;(8)*T_x_* = ∑*L_x_*;(9)*E_x_* = *T_x_*/*N_x_*;(10)*K_x_* = *lnl_x_* − *lnl_x_* + 1(11)
where *x* represents the age class, *a_x_* is the number of individuals in age class *x* after smoothing, and *a*_0_ is the number of individuals in age class I after smoothing; *l_x_* is the number of surviving individuals at the beginning of age class *x*, standardized (usually converted to 1000); *D_x_* is the standardized number of deaths between age class *x* and *x* + 1; *Q_x_* is the mortality rate between age classes *x* and *x* + 1; *L_x_* is the number of individuals still alive during the interval between age classes *x* and *x* + 1, or the interval life expectancy; *T_x_* is the total number of individuals from age class *x* onward; *E_x_* is the expected lifespan or mean life expectancy of individuals entering age class *x*; and *K_x_* is the population extinction rate.

To characterize the survival status of *R. huadingense*, survival curves were plotted based on the three fundamental patterns delineated by Deevey. Type I is a convex curve, indicating that few individuals die before reaching their physiological lifespan; Type II is diagonal, indicating equal mortality at all ages; and Type III is concave, indicating a higher number of early deaths [52]. In this study, we used exponential equations ($N_{x}=N_{0}e^{-bx}$) and power function equations ($N_{x}=N_{0}x^{-b}$) to test and determine the types of survival curves [4], where $N_{x}$ is the number of surviving individuals of diameter class *x* after smoothing, *N*_0_ is the initial number of individuals in the population, and *b* is the mortality rate. The fit was assessed using the coefficient of determination (*R*^2^) and F-test values.

#### 4.3.4. Survival Analysis

To further reveal the structural and dynamic changes in *Rhododendron huadingense* population, a population survival rate $S_{i}$ cumulative mortality rate $F_{i}$, mortality density function $f_{ti}$, and hazard rate $\lambda_{ti}$ were introduced for survival analysis of the population [53], as shown in the following formulas:(12)$$S_{i}=S_{1}\times S_{2}\times S_{3}\cdots \times S_{x}$$(13)$$F_{i}=1-S_{i}$$(14)$$f_{ti}=\frac{S_{i-1}-S_{i}}{h_{i}}$$(15)$$\lambda_{ti}=\frac{2\left(1-S_{i}\right)}{h_{i}\left(1+S_{i}\right)}$$
where *S_i_* represents the survival rate, and *h_i_* denotes the age class width.

### 4.4. Data Processing and Statistical Analysis

All data analysis and visualization were performed using Microsoft Excel 2022, SPSS 27.0.1, R 4.3.2 and Origin Pro 2022.

## Figures and Tables

**Figure 1 plants-14-03406-f001:**
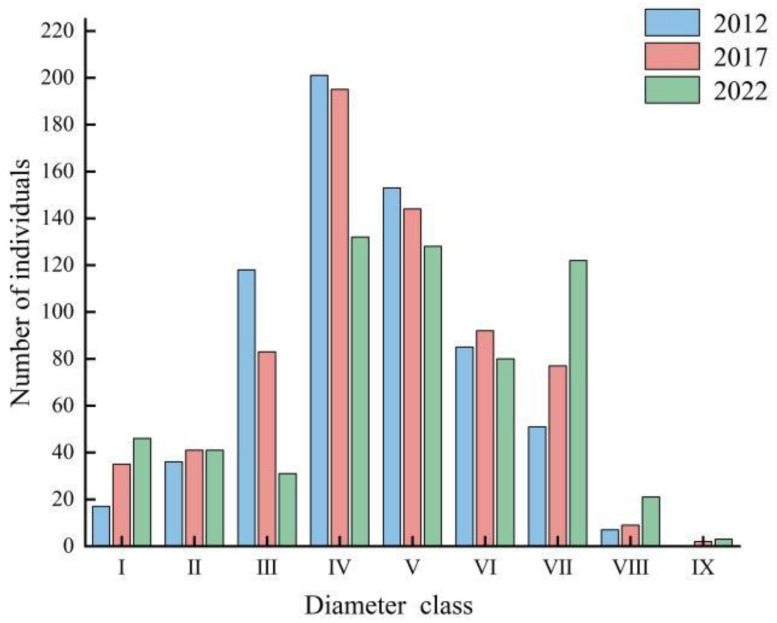
Diameter class structure of *Rhododendron huadingens* population across field surveys conducted in 2012, 2017, and 2022. Note: Diameter at breast height (DBH) was used to define age classes: I (1 cm ≤ DBH < 2 cm), II (2 cm ≤ DBH < 3 cm), III (3 cm ≤ DBH < 4 cm), IV (4 cm ≤ DBH < 5 cm), V (5 cm ≤ DBH < 7 cm), VI (7 cm ≤ DBH < 10 cm), and VII (DBH ≥ 10 cm).

**Figure 2 plants-14-03406-f002:**
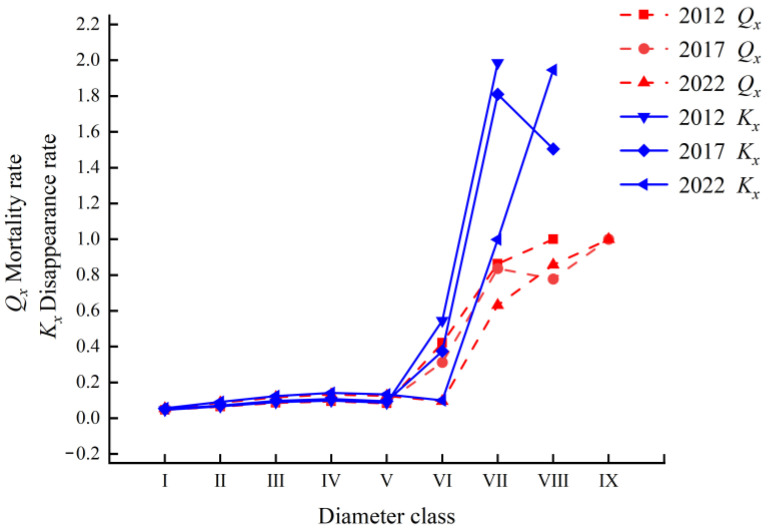
Mortality rate (*Q_x_*) and disappearance rate (*K_x_*) curves of *Rhododendron huadingense* populationin 2012, 2017, and 2022. Diameter at breast height (DBH) was used to define age classes: I (1 cm ≤ DBH < 2 cm), II (2 cm ≤ DBH < 3 cm), III (3 cm ≤ DBH < 4 cm), IV (4 cm ≤ DBH < 5 cm), V (5 cm ≤ DBH < 7 cm), VI (7 cm ≤ DBH < 10 cm), and VII (DBH ≥ 10 cm).

**Figure 3 plants-14-03406-f003:**
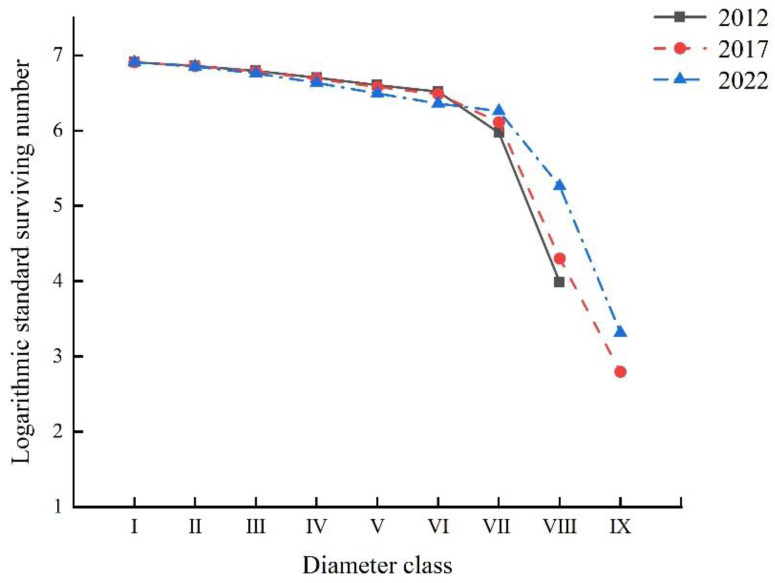
Survival curve of *Rhododendron huadingense* population during census years of 2012, 2017, and 2022. Note: Diameter at breast height (DBH) was used to define age classes: I (1 cm ≤ DBH < 2 cm), II (2 cm ≤ DBH < 3 cm), III (3 cm ≤ DBH < 4 cm), IV (4 cm ≤ DBH < 5 cm), V (5 cm ≤ DBH < 7 cm), VI (7 cm ≤ DBH < 10 cm), and VII (DBH ≥ 10 cm).

**Figure 4 plants-14-03406-f004:**
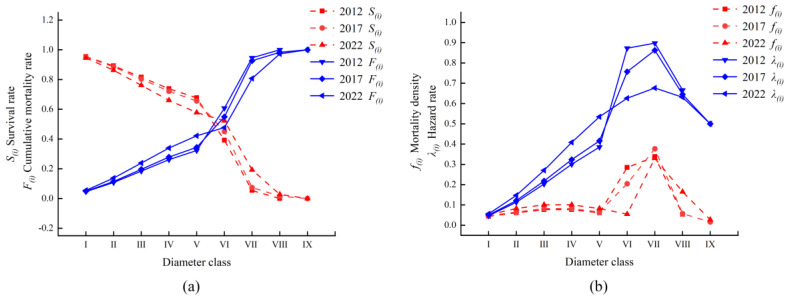
Survival rate (*S_i_*), cumulative mortality rate (*F_i_*), mortality density (*f_i_*), and hazard rate (*λ_i_*) curve of *Rhododendron huadingense* population during census years of 2012, 2017, and 2022. Note: (**a**), Survival rate (*S_i_*) curve of *R. huadingense* population during census years of 2012, 2017, and 2022; (**b**), cumulative mortality rate (*F_i_*) curve of *R. huadingense* population during census years of 2012, 2017, and 2022. Note: Diameter at breast height (DBH) was used to define age classes: I (1 cm ≤ DBH < 2 cm), II (2 cm ≤ DBH < 3 cm), III (3 cm ≤ DBH < 4 cm), IV (4 cm ≤ DBH < 5 cm), V (5 cm ≤ DBH < 7 cm), VI (7 cm ≤ DBH < 10 cm), and VII (DBH ≥ 10 cm).

**Figure 5 plants-14-03406-f005:**
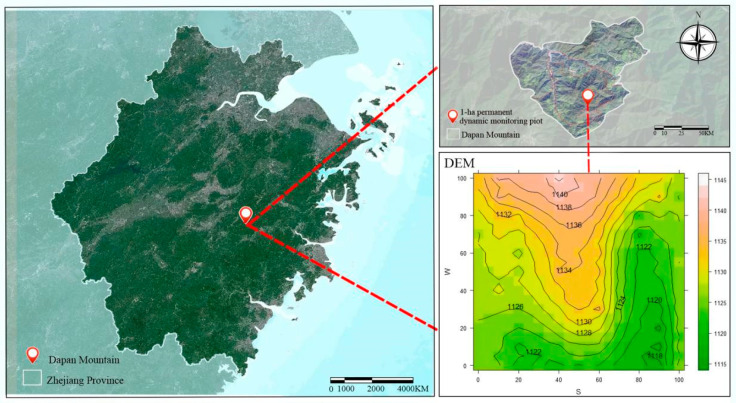
Topographic contour map of 1 hm^2^ fixed monitoring sample plot in Mount Dapan.

**Table 1 plants-14-03406-t001:** Dynamic indices of a *Rhododendron huadingense* population on Mount Dapan.

Diameter Class	Year	*V* _ *n* _								*V* _ *pi* _	*V′* _ *pi* _	*P* _ *max* _
		I	II	III	IV	V	VI	VII	VIII			
Index value	2012	−0.53	−0.7	−0.41	0.24	0.44	0.4	0.86	1	0.177	0.003	0.016
	2017	−0.15	−0.51	−0.57	0.26	0.36	0.16	0.88	0.78	0.179	0.01	0.056
	2022	0.11	0.24	−0.77	0.03	0.38	−0.34	0.83	0.86	0.228	0.008	0.037

Notes: *V_n_*, quantitative dynamic index between age classes *n* and *n* + 1; *V_pi_* and *V′_pi_* represent the quantitative dynamic index of the population without and with considering external interferences, respectively; *P_max_*, maximum risk probability in response to random disturbances. Diameter at breast height (DBH) was used to define age classes: I (1 cm ≤ DBH < 2 cm), II (2 cm ≤ DBH < 3 cm), III (3 cm ≤ DBH < 4 cm), IV (4 cm ≤ DBH < 5 cm), V (5 cm ≤ DBH < 7 cm), VI (7 cm ≤ DBH < 10 cm), and VII (DBH ≥ 10 cm).

**Table 2 plants-14-03406-t002:** Static life table of a *Rhododendron huadingens* population on Mount Dapan.

Year	Diameter Class	*A_x_* (Plant)	*a_x_* (Plant)	*l_x_* (Plant)	*D_x_* (Plant)	*Q_x_*	*L_x_* (Plant)	*T_x_* (Plant)	*E_x_* (Plant)	*S_x_*	*K_x_*	*lnl_x_*
2012	I	25.00	130.00	1000.00	46.15	0.05	976.92	5023.08	5.02	0.95	0.05	6.91
	II	39.00	124.00	953.85	61.54	0.06	923.08	4046.15	4.24	0.94	0.07	6.86
	III	118.00	116.00	892.31	76.92	0.09	853.85	3123.08	3.50	0.91	0.09	6.79
	IV	201.00	106.00	815.38	76.92	0.09	776.92	2269.23	2.78	0.91	0.10	6.70
	V	153.00	96.00	738.46	61.54	0.08	707.69	1492.31	2.02	0.92	0.09	6.60
	VI	85.00	88.00	676.92	284.62	0.42	534.62	784.62	1.16	0.58	0.55	6.52
	VII	51.00	51.00	392.31	338.46	0.86	223.08	250.00	0.64	0.14	1.99	5.97
	VIII	7.00	7.00	53.85	53.85	1.00	26.92	26.92	0.50	0.00	-	3.99
	IX	0.00	0.00	0.00	0.00	-	0.00	0.00	-	-	-	-
2017	I	35.00	122.00	1000.00	49.18	0.05	975.41	5057.38	5.06	0.95	0.05	6.91
	II	41.00	116.00	950.82	65.57	0.07	918.03	4081.97	4.29	0.93	0.07	6.86
	III	83.00	108.00	885.25	81.97	0.09	844.26	3163.93	3.57	0.91	0.10	6.79
	IV	195.00	98.00	803.28	81.97	0.10	762.30	2319.67	2.89	0.90	0.11	6.69
	V	144.00	88.00	721.31	65.57	0.09	688.52	1557.38	2.16	0.91	0.10	6.58
	VI	92.00	80.00	655.74	204.92	0.31	553.28	868.85	1.33	0.69	0.37	6.49
	VII	77.00	55.00	450.82	377.05	0.84	262.30	315.57	0.70	0.16	1.81	6.11
	VIII	9.00	9.00	73.77	57.38	0.78	45.08	53.28	0.72	0.22	1.50	4.30
	IX	2.00	2.00	16.39	16.39	1.00	8.20	8.20	0.50	0.00	-	2.80
2022	I	46.00	109.00	1000.00	55.05	0.06	972.48	5050.46	5.05	0.94	0.06	6.91
	II	41.00	103.00	944.95	82.57	0.09	903.67	4077.98	4.32	0.91	0.09	6.85
	III	31.00	94.00	862.39	100.92	0.12	811.93	3174.31	3.68	0.88	0.12	6.76
	IV	132.00	83.00	761.47	100.92	0.13	711.01	2362.39	3.10	0.87	0.14	6.64
	V	128.00	72.00	660.55	82.57	0.13	619.27	1651.38	2.50	0.88	0.13	6.49
	VI	80.00	63.00	577.98	55.05	0.10	550.46	1032.11	1.79	0.90	0.10	6.36
	VII	122.00	57.00	522.94	330.28	0.63	357.80	481.65	0.92	0.37	1.00	6.26
	VIII	21.00	21.00	192.66	165.14	0.86	110.09	123.85	0.64	0.14	1.95	5.26
	IX	3.00	3.00	27.52	27.52	1.00	13.76	13.76	0.50	0.00	-	3.32

Notes: *A_x_*, actual survival number; *a_x_*, correction value of *A_x_*; *l_x_*, logarithmic standard surviving number; *D_x_*, standardized deaths; *Q_x_*, mortality rate; *L_x_*, surviving individuals of the interval from *x* to *x* + 1; *T_x_*, total number of individuals from level *x* to greater than level *x*; *E_x_*, life expectancy; *S_x_*, survival rate; *K_x_*, extinction rate; *lnl_x_*, natural logarithm of *l_x_*. Diameter at breast height (DBH) was used to define age classes: I (1 cm ≤ DBH < 2 cm), II (2 cm ≤ DBH < 3 cm), III (3 cm ≤ DBH < 4 cm), IV (4 cm ≤ DBH < 5 cm), V (5 cm ≤ DBH < 7 cm), VI (7 cm ≤ DBH < 10 cm), and VII (DBH ≥ 10 cm).

**Table 3 plants-14-03406-t003:** Test models of survival curves of *Rhododendron huadingens* population from 2012 to 2022.

Year	Fitting Equation	*R* ^2^	F	*p*
2012	y = 7.630x − 0.155	0.351	3.241	0.122
	y = 7.977e − 0.056x	0.546	7.213	0.036
2017	y = 8.366x − 0.265	0.385	4.381	0.075
	y = 8.895e − 0.088x	0.61	10.971	0.013
2022	y = 7.927x − 0.2	0.374	4.19	0.08
	y = 8.273e − 0.065x	0.582	9.73	0.017

## Data Availability

No new data were created or analyzed in this study.

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
