# Peer review of "Dynamic Structural Changes in a Population of *Rhododendron huadingense*, a Rare and Endemic Species in Zhejiang, East China"

_plants, 2025, doi:10.3390/plants14223406_

Round 1
Reviewer 1 Report
Comments and Suggestions for Authors
This paper presents interesting research results on the population status, threats, and conservation prospects of a rare shrub species in China – Rhododendron huadingense.
The study covers a period too short – just 10 years – to indicate reliable trends, including the impact of climate change. The decline in the number of juveniles is alarming. At least general information about the species would be useful, such as the height of the individuals, their appearance, their condition, etc. The lifespan of the oldest individuals is unknown. The authors estimate the age of individuals based on shoot thickness, but it is known that growth rates can vary between individuals depending on many factors, including climatic factors such as humidity, temperature, and soil type. Therefore, determining the age of individuals based on shoot thickness remains highly controversial. If the age of a plant corresponding to a specific shoot thickness were determined, it would be possible to discuss its age structure. Without this research, describing the age of the plant is unjustified. An older plant may have a thinner shoot than a younger one because it grew in unfavorable conditions. Was the average annual growth rate for a specific specimen examined? Was the thickest shoot selected? Were the exact same specimens examined in specific years?
Article layout – the research area and methods are typically presented before the results, rather than at the end, as here.
Author Response
|
Response 1: We sincerely appreciate your valuable feedback. Regarding your observation of the "short research duration," we have conducted thorough analysis. To clarify the significance and rationale of the 10-year time span in this study, we provide the following explanations: 1. Life Cycle and Regeneration Characteristics of Rhododendron huadingicum is a deciduous shrub or small tree species that grows slowly under natural conditions. Five-year-old seedlings typically reach a height of approximately 50 cm, while mature plants require multiple years to flower and produce fruit. The 10-year period covers critical growth stages from seedling to maturity, providing sufficient observation time to monitor population dynamics such as the proportion of young plants and natural regeneration rates. 2. Timeliness in Endangered Species Conservation Research: The R. huadingense, classified as a nationally protected second-class wild plant with an endangered status, has fewer than 1,000 wild individuals remaining. For such critically small populations, prioritizing short-term monitoring data is essential. Ten-year monitoring programs can effectively track initial outcomes of conservation interventions (including habitat restoration and reintroduction efforts), providing critical data to optimize protection strategies. 3. Supplementary and Innovative Design of the study This study is the first project to systematically monitor the population of R. huadingense for ten years. The sample plots, methods and baseline data established will provide a comparative framework for subsequent follow-up studies and promote long-term ecological research on this species. 4. Justifying the time span through practical needs. As evidenced by numerous endangered species studies: Many research on endangered plants (e.g., Acer miaotaiense(2018-2022) and Rhododendron rex subsp. Rex(2017-2019)) publish key findings based on less than 5 years of monitoring data, emphasizing the correlation between data quality and ecological processes rather than the mere duration. |
|
Comments 2: At least general information about the species would be useful, such as the height of the individuals, their appearance, their condition, etc. |
|
Response 2: Thank you for pointing this out. I Agree. We have, accordingly, done modified “Rhododendron huadingense is a deciduous shrub 1–4 m tall. Bark gray-brown, with prominent longitudinal fissures. Leaves membranous to thinly chartaceous, 4-5-verticillate at branch apex; blade elliptic, margin finely serrulate with teeth gland-tipped or shortly ciliate. Inflorescence a terminal umbel-like raceme, 2-4-flowered. Corolla funnel-campanulate, pale purplish-red to purplish-red. Exhibiting hysteranthous flowering with leaf buds undeveloped at anthesis, presenting an elegant and archaic appearance with high ornamental value. However, this species is critically endangered due to extremely narrow distribution, small population size, and severe threats from illegal collection.” to emphasize this point. The morphological characteristics of R. huadingense have been added, with the revised section located in the second paragraph of the introduction on page 2, lines 3-10. |
|
Comments 3: The authors estimate the age of individuals based on shoot thickness, but it is known that growth rates can vary between individuals depending on many factors, including climatic factors such as humidity, temperature, and soil type. Therefore, determining the age of individuals based on shoot thickness remains highly controversial. If the age of a plant corresponding to a specific shoot thickness were determined, it would be possible to discuss its age structure. Without this research, describing the age of the plant is unjustified. An older plant may have a thinner shoot than a younger one because it grew in unfavorable conditions. |
|
Response 3: We sincerely appreciate your valuable feedback. We fully understand and endorse your insightful observation that "variations in growth rates create uncertainties when determining tree age based solely on trunk diameter." This is precisely the critical issue requiring careful consideration in tree population ecology research. Below, we will elaborate on the rationale for using diameter-class structure analysis by integrating ecological principles with our study design, aiming to address your concerns. The use of diameter-based instead of age-based structural analysis is a common ecological approach known as "spatial substitution for temporal measurement." This method serves as a classic tool for studying long-term ecological dynamics. Its core premise is that individuals of varying sizes represent different developmental stages of a population. Generally, increased trunk diameter correlates positively with a tree's growth years. 1. The method's extensive practical applications: This approach has been widely applied in studies of various species. Within the same ecological environment, the response patterns of diameter at breast height (DBH) and age classes of the same tree species to environmental factors show consistent characteristics. Research demonstrates a significant positive correlation between DBH size and age in species such as R. huadingense. For example, in species like Pteroceltis tatarinowii, Ficus microcarpa, Ficus chinensis, Phoebe zhennan, and Quercus liaodongensis, DBH measurements are typically divided into discrete diameter classes at regular intervals, with each class corresponding to a specific age bracket. For shrubs or small trees like R. huadingense, this approach can be adapted by measuring diameter classes and establishing regression models to estimate age. 2. Considering factors such as climate (including humidity, temperature, and soil type), we integrated other age assessment indicators—such as tree height, crown width, and branching patterns—into our analysis. These morphological metrics served as supplementary validation tools for diameter class determination, enabling a more comprehensive evaluation of population structure dynamics. 3. Rhododendron huadingense is a key protected wild plant in Zhejiang Province, making it unsuitable to determine age by extracting tree cores for ring counting or through stem analysis. Such methods are prohibited or require extreme caution, especially for rare and endangered species. In contrast, measuring DBH is rapid, non-destructive, and can be conducted on a large scale. 4.Robustness of Population Pattern Conclusions: While individual age estimates may contain some errors, the overall hierarchical distribution pattern revealed by the large-scale 1-ha² sample (comprising 668 surveyed individuals) – including the distinct inverted-J distribution and seedling pool depletion observed in this study – demonstrates significant stability. This comprehensive pattern reflects the long-term combined effects of birth rates, mortality rates, and growth processes, providing reliable indicators of population renewal patterns and developmental dynamics. |
|
Comments 4: Were the exact same specimens examined in specific years? |
|
Response 4: Regarding your inquiry about whether identical samples were tested in specific years, we hereby clarify: This study conducted continuous monitoring of Rhododendron huadingsii populations over a 10-year period (2012–2022) within a fixed one-hectare monitoring plot. The core feature of this approach lies in the repeated measurement and tracking of 'identical samples' (all individuals within the plot). Below is a detailed description of our study design, which aims to clarify how we ensure sample consistency and data comparability. Fixed sampling and individual tracking are the core designs to achieve long-term and accurate exploration of population dynamics. In this study, fixed sampling and individual tracking methods were used. 1. Determination of Sampling Scope: The core of this study lies in establishing a permanent fixed plot of 1 hectare (100 m × 100 m) within the Dapingshan National Nature Reserve. This clearly defined and unchanging geographical boundary encompasses all individuals of Rhododendron huadingense under investigation. 2. Individual Identification and Tagging: In 2012, researchers conducted plaque installation, numbering, mapping, and documentation for every R. huadingence plant within the plot. This ensured each individual received a unique identifier, while meticulously recording initial characteristics including location, basal diameter, tree height, and crown spread. 3. Plot Reassessment: Follow-up surveys were conducted in 2017 and 2022, during which newly added trees (with trunk diameter ≥1.0 cm) were tagged and numbered, while dead specimens were documented. This approach ensured the collected data formed longitudinal time-series observations of the same cohort, fully meeting the requirement of "testing identical samples." |
|
Comments 5:Was the average annual growth rate for a specific specimen examined? |
|
Response5: We sincerely appreciate your insightful inquiry. Your focus on "specific sample annual growth rates" directly addresses the core of population dynamics research. Here we provide detailed explanations: Our study did not calculate "annual growth rates" for specific samples, primarily due to four key considerations: 1.This study aimed to evaluate the overall population dynamics and structural stability, not individual growth rates. We therefore applied the widely accepted static life table approach ("substituting space for time") from conservation ecology. 2. Growth Trends Captured by Population-Level Indicators These indices integrate information across age classes, reflecting birth, death, and growth potential collectively. 3. Alignment with Traditional Growth Rates 4. Advantages of the Approach |
|
Comments 6: The lifespan of the oldest individuals is unknown.Was the thickest shoot selected? |
|
Response 6: Thank you for raising this important question. In population ecology research, scholars commonly use indirect methods based on morphological indicators to assess the relative age and developmental stages of individuals. The scientific rationale for selecting the "thickest branch" (i.e., the individual with the largest basal diameter) is based on the fact that, in our study, the individual with the "largest basal diameter" in a plot is considered potentially the oldest or at least the most developmentally mature representative within that specific plot. |
|
Comments 7: Article layout – the research area and methods are typically presented before the results, rather than at the end, as here. |
|
4. Response to Comments on the Quality of English Language We have ask Mr. Philip Haytt, an English native-speaker who worked in Forestry Depart in USA, to help us and improve the English again. |
Reviewer 2 Report
Comments and Suggestions for Authors
Rewrite the introduction in a more formal academic style suitable for a scientific journal.
The manuscript would benefit from a clearer contextualization of the topic by highlighting relevant previous studies. In addition, the study aims are currently presented in rather general terms and could be more explicitly detailed to strengthen the focus and clarity of the research.
Results: It appears that the volume of data is substantial. However, it is not presented in an appropriate manner. Statistical analysis is required to validate the results obtained.
Materials and methods: Add reference to Figure 5, the map, for better understanding.
Add a relevant figure for establishing the plot and the population survey.
The discussions are generally well written. However, after reviewing the results, they may change.
Author Response
|
Comments 1: Rewrite the introduction in a more formal academic style suitable for a scientific journal. |
|
Response 1: We are grateful for the valuable suggestions provided by the reviewers. In response to their feedback, we have thoroughly revised the introduction section: We have further standardized the language style to align with academic journal publication requirements; enhanced content organization by supplementing existing literature related to the species under study, thereby establishing a solid foundation for the research context; and explicitly outlined three core scientific questions addressed in this paper at the conclusion of the introduction, providing clear guidance for the subsequent discussion. |
|
Comments 2: The manuscript would benefit from a clearer contextualization of the topic by highlighting relevant previous studies. In addition, the study aims are currently presented in rather general terms and could be more explicitly detailed to strengthen the focus and clarity of the research. |
|
Response 2: Thank you very much for your valuable comments on our manuscript. n response to their feedback, we have revised the introduction section: We have further standardized the language style to align with academic journal publication requirements; enhanced content organization by supplementing existing literature related to the species under study, “Many scholars have conducted preliminary studies on R. huadingense, with research efforts primarily focused on areas such as resource surveys, community characteristics, genetic diversity, and artificial propagation techniques. However, existing studies have largely emphasized static descriptions of population characteristics and explorations of conservation techniques, while systematic monitoring and investigation of the dynamic patterns and driving mechanisms of its wild populations across different successional stages or over long-term temporal scales remain notably lacking.”[Pages 2, paragraph 3 lines 12-19.] thereby establishing a solid foundation for the research context. And explicitly outlined three core scientific questions addressed in this paper at the conclusion of the introduction, providing clear guidance for the subsequent discussion.“Therefore, this study focuses on the rare and endangered species R. huadingense. Field investigations were conducted to examine the population regeneration characteristics and distribution patterns within its natural range. By analyzing the age structure, static life table, and survival curve of wild populations, the study aims to address the following scientific questions: (1) the survival status of the population over the past decade; (2) the dynamic trends of the population; (3) the mechanisms underlying its endangerment and conservation. The findings will elucidate the population maintenance mechanisms and endangerment causes from an ecological perspective, while providing a scientific basis for developing effective conservation strategies and enabling targeted protection of this species.”[Pages 3, paragraph2, lines 9-18] |
|
Comments 3: Results: It appears that the volume of data is substantial. However, it is not presented in an appropriate manner. Statistical analysis is required to validate the results obtained. |
|
Response 3: We sincerely appreciate the reviewers' valuable feedback. We fully recognize that rigorous statistical analysis is essential for ensuring the reliability of research conclusions. To clarify, this study has incorporated standardized statistical methods throughout the entire process of chart creation and result interpretation. Specifically: In the population structure analysis (Figure 1), we established age-class structures based on DBH, which provided a foundation for subsequent analyses. 1. In the fitting of the survival curve (Figure 3), we not only plotted the standardized survival curve but also compared two models: the exponential and the power function. The statistical results indicated that the exponential model demonstrated greater explanatory power, leading us to classify this population as Type Deevey-II. 2. In the population dynamics prediction, we further calculated the population dynamics index as a quantitative indicator of the decline risk, and used the first moving average method to predict the size of the next three age classes (Table 1). The statistical results all pointed to the conclusion that the population will continue to grow. In conclusion, all our conclusions, including "insufficient replenishment of young individuals in the population and obstacles to renewal", are derived from the statistical analysis of the above system. We have clarified the relevant analysis methods in the revised draft, and thank the reviewers for their review. |
|
Comments 4: Materials and methods: Add reference to Figure 5, the map, for better understanding. Add a relevant figure for establishing the plot and the population survey. |
|
Response 4: Thank you for the constructive guidance from the reviewers regarding the Materials and Methods section. Based on their feedback, we have made the following improvements: 1. Concerning data visualization processes, the explanatory diagrams were submitted as supplementary materials during submission for reference by reviewers and readers. 2. Additionally, we have clearly listed all software tools and version information used for chart creation in “Section 4.4 Data Processing and Statistical Analysis of the Methods section”, ensuring reproducibility throughout the research process.[Pages 15,, paragraph2, lines 1] |
|
Comments 5: The discussions are generally well written. However, after reviewing the results, they may change. |
|
Response 5: We thank the reviewers for the preliminary approval of some parts of the discussion and their careful attention to the overall logic of the paper. We fully understand and endorse your perspective. The discussion section must strictly adhere to the research findings presented in this paper. We are fully prepared to conduct a systematic review and make necessary adjustments to the discussion section immediately after completing all data modifications, chart revisions, and statistical analyses based on your feedback regarding the "Results" section. This ensures that all inferences and interpretations in the discussion align precisely with the final results and maintain logical consistency. |
|
4. Response to Comments on the Quality of English Language We have ask Mr Philip Haytt, an English native speaker who worked in Forestry Department USA, to help us and improve the English again. |
Reviewer 3 Report
Comments and Suggestions for Authors
Dear authors this is a very interesting work for the conservation of the endemic species Rhododendron huadingense.
These are my comments and suggestions
Keywords: please do not use words already mentioned in the title e.g. Rhododendron huadingense
Please make sure that all scientific names are italicized
The title is perfectly clear.
The abstract summarises the objectives, the methods and the key results. Some statements (e.g. 'growth potential under optimal conditions') could be simplified or clarified. Please consider reducing the length of the abstract to approximately 250 words.
The introduction highlights the importance of demographic studies for endemic species and provides good information on the distribution, habitat and threats to R. huadingense. However several sentences restate similar ideas (e.g. rarity, fragmentation and anthropogenic threats).
Results: Please make sure that the figure legends define all abbreviations and consider moving large data tables, such as the static life table, to the supplementary materials.
The discussion provides a solid ecological interpretation and a literature comparison. However it repeats data from the 'Results' section and the sentences are sometimes long.
Materials and Methods: The text would benefit from brief explanations of variables (for example, the meaning of 'Sn' and 'K' should be clarified).
Figures, tables and references: The figures and tables are informative. References are thorough and up to date. There are minor inconsistencies in the citation format (e.g. missing full stops after initials).
Comments on the Quality of English LanguageThere are repeated phrases, such as 'The population exhibits a certain sensitivity to stochastic disturbances', which appears twice. There is overuse of the passive voice. English editing by a native spealer is recommended.
Author Response
|
Comments 1: Keywords: please do not use words already mentioned in the title e.g.Rhododendron huadingense |
|
Response 1:We appreciate the valuable suggestions from the reviewers. We have followed your suggestions and deleted elevant redundant words in keywords. |
|
Comments 2: Please make sure that all scientific names are italicized The title is perfectly clear. |
|
Response 2: We appreciate the reviewers valuable feedback. Following their suggestions, we have conducted a thorough review of the entire text to ensure all biological terms are properly italicized. Additionally, we have refined the article's title to better capture the essence of the research. Rhododendron huadingense and R. huadingense throughout the entire text. |
|
Comments 3: The abstract summarises the objectives, the methods and the key results. Some statements (e.g. 'growth potential under optimal conditions') could be simplified or clarified. Please consider reducing the length of the abstract to approximately 250 words. |
|
Response 3: We appreciate the detailed and constructive feedback provided by the reviewers on the abstract. We have carefully revised it based on your suggestions. 1. Language refinement: For expressions such as "growth potential under optimal conditions", we have deleted "R. huadingense", "under optimal conditions", "success", "to reduce environmental stressors", and "These conservation practices can effectively promote the long-term persistence of R. huadingense". We have opted to directly state the relevant specific research findings or conclusions, thus avoiding ambiguity. |
|
Comments 4: The introduction highlights the importance of demographic studies for endemic species and provides good information on the distribution, habitat and threats to R. huadingense. However several sentences restate similar ideas (e.g. rarity, fragmentation and anthropogenic threats). |
|
Response 4: We appreciate the detailed review and valuable feedback provided by the reviewers on the introduction. We have fully acknowledged the issue of viewpoint repetition in the original text, particularly the repeated references to species rarity, habitat fragmentation, and human threats. We have thoroughly reviewed and streamlined the introduction section, eliminating redundant expressions and consolidating relevant arguments to ensure a logical progression of core viewpoints. |
|
Comments 5: Results: Please make sure that the figure legends define all abbreviations and consider moving large data tables, such as the static life table, to the supplementary materials. |
|
Response 5: Thank you for your review comments. We have made the following revisions to the manuscript: All figure captions have been thoroughly reviewed and enhanced to ensure clear labeling of abbreviations. Regarding the static life tables, we have streamlined and restructured their content, retaining core data in the main text tables. The complete version remains accessible in the main text for readers to directly consult key data. All the above modifications have been reflected in the manuscript. Please review. |
|
Comments 6: The discussion provides a solid ecological interpretation and a literature comparison. However it repeats data from the 'Results' section and the sentences are sometimes long. |
|
Response 6:We sincerely appreciate your thorough review of the manuscript and the valuable feedback you provided. We have carefully revised the text based on your suggestions in 'Results' section, condensing repetitive sections as follows: “Static life tables, survival curves, and survival function analysis can reflect a population's survival status and environmental impact degree, and predict its future trends[25-27]. In this study, static life tables from three surveys show that the standardized survival number (lx) and life expectancy (Ex) of the studied Rhododendron huadingense population decline with increasing diameter classes and indicating the population is approaching its physiological lifespan. This pattern aligns consistently with the species' biological characteristics. The mortality rate (Qx) and disappearance rate (Kx) curves of the R. huadingense population showed generally consistent trends, both exhibiting a continuous increase with diameter class and reaching their maximum values in diameter classes VI‒IX. Both life expectancy (Ex) and survival rate showed significant declines after diameter class VI、VII, suggesting that increasing diameter class leads to growth resources demands and intensified competition among individuals. These findings are consistent with Wang’s analysis of community structure and population dynamics in a Rhododendron xishuiense population from Guizhou[28].”[Pages 10, paragraph2, lines 1] |
|
Comments 7: Materials and Methods: The text would benefit from brief explanations of variables (for example, the meaning of 'Sn' and 'K' should be clarified). |
|
Response 7: Thank you for your review comments. We have made the following revisions to the manuscript: add“K denotes the total number of age classes in the population, and Sn represents the number of individuals in the n-th age class.”in Pages 13, paragraph3, lines 6. |
|
Comments 8: Figures, tables and references: The figures and tables are informative. References are thorough and up to date. There are minor inconsistencies in the citation format (e.g. missing full stops after initials |
|
Response 8 We sincerely appreciate your thorough review of this manuscript and the valuable suggestions for improvement. We fully acknowledge your observations regarding minor inconsistencies in citation formatting, and have conducted a comprehensive review and revision of the entire text in accordance with academic standards. For instance, we have corrected 'Wang J.' to 'Wang, J.'',Xu, X.F to Xu, X.F and Zhou, L.' to 'Zhou, L. in Pages 15 “1”,”12”,”40” |
|
4. Response to Comments on the Quality of English Language |
|
Point 1: There are repeated phrases, such as 'The population exhibits a certain sensitivity to stochastic disturbances', which appears twice. There is overuse of the passive voice. English editing by a native spealer is recommended. |
|
Response 1: Thank you for your suggestion. I have deleted the repeated sentences and will invited native speakers to revise the sentence. |
We have ask Mr Philip Haytt, an English native speaker who worked in Forestry Department USA, to help us and improve the English again.
Round 2
Reviewer 2 Report
Comments and Suggestions for Authors
Add more references.
Anyway, the manuscript has been improved. The authors have responded to the reviewer's comments. I consider that it satisfies the standards required for publication.
Author Response
Comments 1: Add more references.
Response 1: Yes, we really accepted the comments and added several relative references in Introduction and Discussion. We have changed the literature numbers in order.
Comments 2: Anyway, the manuscript has been improved. The authors have responded to the reviewer's comments. I consider that it satisfies the standards required for publication.
Response 2: no changes need.
Reviewer 3 Report
Comments and Suggestions for Authors
Dear authors I have only some comments with suggestions for minor corrections
Please write the title as follows: Dynamic structural changes in a population of Rhododendron huadingense, a rare and endemic species in Zhejiang, East China
Please don't use words already mentioned in the title as keywords for instance Rhododendron huadingense
Author Response
Comments 1: Please write the title as follows: Dynamic structural changes in a population of Rhododendron huadingense, a rare and endemic species in Zhejiang, East China
Response 1: this title was just used in our manuscript.
Comments 2: Please don't use words already mentioned in the title as keywords for instance Rhododendron huadingense
Response 2: Yes, the key words were different from the title except for Rhododendron huadingense